

# Finding the optimal Bayesian network given a constraint graph

Jacob M. Schreiber[1] and William S. Noble[2]

[1] Department of Computer Science, University of Washington, Seattle, WA, United States of America
[2] Department of Genome Science, University of Washington, Seattle, WA, United States of America

## ABSTRACT

Despite recent algorithmic improvements, learning the optimal structure of a Bayesian network from data is typically infeasible past a few dozen variables. Fortunately, domain knowledge can frequently be exploited to achieve dramatic computational savings, and in many cases domain knowledge can even make structure learning tractable. Several methods have previously been described for representing this type of structural prior knowledge, including global orderings, super-structures, and constraint rules. While super-structures and constraint rules are flexible in terms of what prior knowledge they can encode, they achieve savings in memory and computational time simply by avoiding considering invalid graphs. We introduce the concept of a "constraint graph" as an intuitive method for incorporating rich prior knowledge into the structure learning task. We describe how this graph can be used to reduce the memory cost and computational time required to find the optimal graph subject to the encoded constraints, beyond merely eliminating invalid graphs. In particular, we show that a constraint graph can break the structure learning task into independent subproblems even in the presence of cyclic prior knowledge. These subproblems are well suited to being solved in parallel on a single machine or distributed across many machines without excessive communication cost.

# INTRODUCTION

Bayesian networks are directed acyclic graphs (DAGs) in which nodes correspond to random variables and directed edges represent dependencies between these variables. Conditional independence between a pair of variables is represented as the lack of an edge between the two corresponding nodes. The parameters of a Bayesian network are typically simple to interpret, making such networks highly desirable in a wide variety of application domains that require model transparancy.

Frequently, one does not know the structure of the Bayesian network beforehand, making it necessary to learn the structure directly from data. The most intuitive approach to the task of Bayesian network structure learning (BNSL) is "search-and-score," in which one iterates over all possible DAGs and chooses the one that optimizes a given scoring function. Recent work has described methods that find the optimal Bayesian network structure without explicitly considering all possible DAGs (*Malone, Yuan & Hansen, 2011*; *Yuan, Malone &*

Corresponding author
Jacob M. Schreiber,
jmschr@cs.washington.edu

*Wu, 2011*; *Fan, Malone & Yuan, 2014*; *Jaakkola et al., 2003*), but these methods are still infeasible for more than a few dozen variables. In practice, a wide variety of heuristics are often employed for larger datasets. These algorithms, which include branch-and-bound (*Suzuki, 1996*), Chow-Liu trees (*Chow & Liu, 1968*), optimal reinsertion (*Moore & Wong, 2003*), and hill-climbing (*Tsamardinos, Brown & Aliferis, 2006*), typically attempt to efficiently identify a structure that captures the majority of important dependencies.

In many applications, the search space of possible network structures can be reduced by taking into account domain-specific prior knowledge (*Gamberoni et al., 2005*; *Zuo & Kita, 2012*; *Schneiderman, 2004*; *Zhou & Sakane, 2003*). A simple method is to specify an ordering on the variables and require that parents of a variable must precede it in the ordering (*Cooper & Herskovits, 1992*). This representation leads to tractable structure learning because identifying the parent set for each variable can be carried out independently from the other variables. Unfortunately, prior knowledge is typically more ambiguous than knowing a full topological ordering and may only exist for some of the variables. A more general approach to handling prior knowledge is to employ a "super-structure," i.e., an undirected graph that defines the super-set of edges defining valid learned structures, forbidding all others (*Perrier, Imoto & Miyano, 2008*). This method has been fairly well studied and can also be used as a heuristic if defined through statistical tests instead of prior knowledge. A natural extension of the undirected super-structure is the directed super-structure (*Ordyniak & Szeider, 2013*), but to our knowledge the only work done on directed super-structures proved that an acyclic directed super-structure is solvable in polynomial time. An alternate, but similar, concept is to define which edges must or cannot exist as a set of rules (*Campos & Ji, 2011*). However, these rule-based techniques do not specify how one would exploit the constraints to reduce the computational time past simply skipping over invalid graphs.

We propose the idea of a "constraint graph" as a method for incorporating prior information into the BNSL task. A constraint graph is a directed graph where each node represents a set of variables in the BNSL problem and edges represent which variables are candidate parents for which other variables. The primary advantage of constraint graphs versus other methods is that the structure of the constraint graph can be used to achieve savings in both memory cost and computational time beyond simply eliminating invalid structures. This is done by breaking the problem into independent subproblems even in the presence of cyclic prior knowledge. An example of this cyclic prior knowledge is identifying two groups of variables that can draw parents only from each other, similar to a biparte graph. It can be difficult to identify the best parents for each variable that does not result in a cycle in the learned structure. In addition, constraint graphs are visually more intuitive than a set of written rules while also typically being simpler than a super-structure, because constraint graphs are defined over sets of variables instead of the original variables themselves. This intuition, combined with automatic methods for identifying parallelizable subproblems, makes constraint graphs easy for non-experts to define and use without requiring them to know the details of the structure learning task. This technique is similar to work done by *Fan, Malone & Yuan (2014)*, where the authors describe the same computational gains through the identification of "potentially optimal

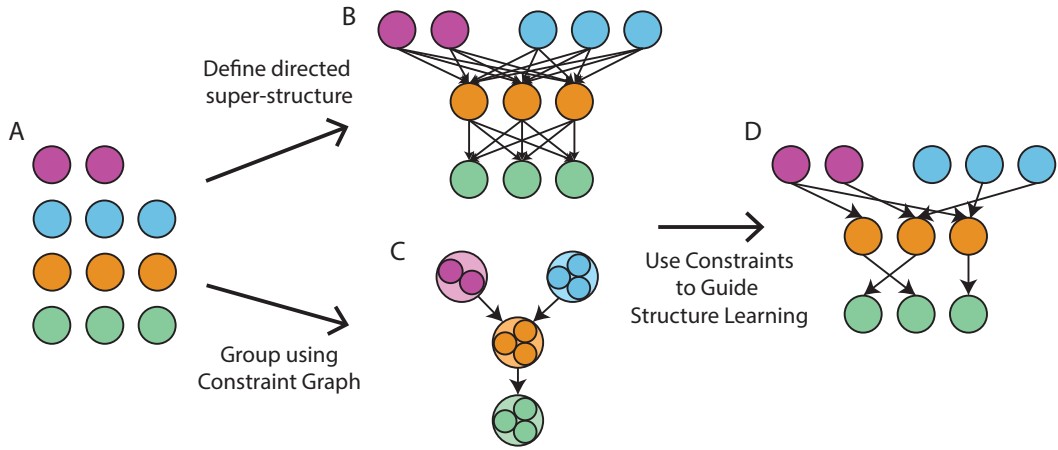

**Figure 1** **A constraint graph grouping variables.** (A) We wish to learn a Bayesian network over 11 variables. The variables are colored according to the group that they belong to, which is defined by the user. These variables can either (B) be organized into a directed super structure or (C) grouped into a constraint graph to encode equivalent prior knowledge. Both graphs define the superset of edges which can exist, but the constraint graph uses far fewer nodes and edges to encode this knowledge. (D) Either technique can then be used to guide the BNSL task to learn the optimal Bayesian network given the constraints.

parent sets.'' One difference is that Fan et al. define the constraints on individual variables instead of on sets on variables, as this work does. By defining the constraints on sets of variables instead of individual ones, one can identify further computational gains when presented with cyclic prior knowledge. Given that two types of graphs will be discussed throughout this paper, the Bayesian network we are attempting to learn and the constraint graph, we will use the terminology ''variable'' exclusively in reference to the Bayesian network and ''node'' exclusively in reference to the constraint graph.

## CONSTRAINT GRAPHS

A constraint graph is a directed graph in which nodes contain disjoint sets of variables from the BNSL task, and edges indicate which sets of variables can serve as parents to which other sets of variables. A self-loop in the constraint graph indicates that no prior knowledge is known about the relationship between variables in that node, whereas a lack of a self-loop indicates that no variables in that particular node can serve as parents for another variable in that node. Thus, the naive BNSL task can be represented as a constraint graph consisting of a single node with a self-loop. A constraint graph can be thought of as a way to group the variables (Fig. 1A), define relationships between these groups (Fig. 1C), and then guide the BNSL task to efficiently find the optimal structure given these constraints (Fig. 1D). In contast, a directed super-structure defines all possible edges that can exist in accordance with the prior knowledge (Fig. 1B). Typically, a directed super-structure is far more complicated than the equivalent constraint graph. Cyclic prior knowledge can be represented as a simple cycle in the constraint graph, such that the variables in node $A$ draw their parents solely from node $B$, and $B$ from $A$.

Any method for reducing computational time through prior knowledge exploits the "global parameter independence property" of BNSL. Briefly, this property states that the optimal parents for a variable are independent of the optimal parents for another variable given that the variables do not form a cycle in the resulting Bayesian network. This acyclicity requirement is typically computationally challenging to determine because a cycle can involve more variables than the ones being directly considered, such as a graph which is simply a directed loop over all variables. However, given an acyclic constraint graph or an acyclic directed super-structure, it is impossible to form a cycle in the resulting structure; hence, the optimal parent set for each variable can be identified independently from all other variables. A convenient property of constraint graphs, and one of their advantages relative to other methods, is that independent subproblems can be found through global parameter independence even in constraint graphs which contain cycles. We describe in 'Solving a component of the constraint graph' the exact algorithm for finding optimal parent sets for each case one can encounter in a constraint graph. Briefly, the constraint graph is first broken up into its strongly connected components (SCCs) that identify which variables can have their parent sets found independently from all other variables ("solving a component") without the possibility of forming a cycle in the resulting graph. Typically these SCCs will be single nodes from the constraint graph, but may be comprised of multiple nodes if cyclic prior knowledge is being represented. In the case of an acyclic constraint graph, all SCCs will be single nodes, and in fact each variable can be optimized without needing to consider other variables, in line with theoretical results from *Ordyniak & Szeider (2013)*. In addition to allowing these problems to be solved in parallel, this breakdown suggests a more efficient method of sharding the data in a distributed learning context. Specifically, one can assign an entire SCC of the constraint graph to a machine, including all columns of data corresponding to the variables in that SCC and all variables in nodes which are parents to nodes in the SCC. Given that all subproblems which involve this shard of the data are contained in this SCC of the constraint graph, there will never be duplicate shards and all tasks involving a shard are limited to the same machine. The concept of identifying SCCs as independent subproblems has also been described in *Fan, Malone & Yuan (2014)*.

It is possible to convert any directed super-structure into a constraint graph and vice-versa though it is far simpler to go from a constraint graph to a directed super-structure. To convert from a directed super-structure to a constraint graph, one must first identify all strongly connected components that are more than a single variable. All variables in a strongly connected component can be put into the same node in a constraint graph that contains a self loop. Then, one would tabulate the unique parent and children sets a variable can have. All variables outside of the previously identified strongly connected components with the same parent and children sets can be grouped together into a node in the constraint graph. Edges then connect these sets based on the shared parent sets specified for each node. In the situation where a node in the constraint graph can draw parents from only a subset of the variables in a node created by the identification of the strongly connected components, the node must be broken into two nodes that both have self loops and loops connecting to each other to allow for only a subset of those variables

to serve as a parent for another node. In contrast, to convert from a constraint graph to a directed super-structure one would simply draw, for each node, an edge from all variables in the current node to all variables in the node's children. We suggest that constraint graphs are the more intuitive method both due to their simpler representation and ease of extracting computational benefits from the task.

## METHODS

### Bayesian network structure learning

Although solving a component in a constraint graph can be accomplished by a variety of algorithms including heuristic algorithms, we assume for this paper that one is using some variant of the exact dynamic programming algorithm proposed by *Malone, Yuan & Hansen (2011)*. We briefly review that algorithm here.

The goal of the algorithm is to identify the optimal Bayesian network defined over the set of variables without having to repeat any calculations and without having to use excessive memory. This is done by defining additional graphs, the parent graphs and the order graph. We will refer to each node in these graphs as "entries" to distinguish them from the constraint graph and the learned Bayesian network. A parent graph is defined for each variable and can be defined as a lattice, where the entries to some layer $i$ correspond to combinations of all other variables of size $i$. Each entry is connected to the entries in the previous layers that are subsets of that entry such that $(X_1, X_2)$ would be connected to both $X_1$ and $X_2$. For each entry, the score of the variable is calculated using the parents in the entry and compared to the scores held in the parent entries, recording only the best scoring value and parent set amongst them. These entries then hold the dynamically calculated best parent set and associated score, allowing for constant time lookups later on the best parent set given a set of possible parents. The order graph is structured in the same manner as the parent graphs except over all variables. In contrast with the parent graphs, it is the edges that store useful information in the form of the score associated with adding a given variable to the set of seen variables stored in the entry and the parent set that yields this score. Each path from the empty root node to the leaf node containing the full set of variables encodes the optimal network given a topological sort of the variables, and the shortest path encodes the optimal network. This data structure reduces the time required to find the optimal Bayesian network from $O(n2^{n(n-1)})$ time in the number of variables to $O(n2^n)$ time in the number of variables without the need to keep a large cache of values.

Structure learning is flexible with respect to the score function used to identify the optimal graph. There are many score functions that typically aim to penalize the log likelihood of the data by the complexity of the graph to encourage sparser structures. These usually come in the form of Bayesian score functions, such as Bayesian-Dirichlet (*Heckerman, Geiger & Chickering, 1995*), or those derived from information theory, such as minimum description length (MDL) (*Suzuki, 1996*). Most score functions decompose across variables of a Bayesian network according to the global parameter independence property, such that the score for a dataset given a model is equal to the product of the score of each variable given its parents. While constraint graphs remain agnostic to the specific

score function used, we assume that MDL is used as it has several desirable computational benefits. For review, MDL defines the score as the following:

$$MDL(D|M) = P(D|M) - \frac{1}{2}\log(N)|B| \qquad (1)$$

where $|B|$ defines the number of parameters in the network. The term "minimum description length" arises from needing $\frac{1}{2}\log(N)$ bits to represent each parameter in the model, making the second term the total number of bits needed to represent the full model. The MDL score function has the convenient property that a variable cannot have more than $\log\left(\frac{n}{\log(n)}\right)$ parents given $n$ samples, greatly reducing computational time.

## Solving a component of the constraint graph

The strongly connected components of a constraint graph can be identified using Tarjan's algorithm (*Tarjan, 1971*). Each SCC corresponds to a subproblem of the constraint graph and can be solved independently. In many cases the SCC will be a single node of the constraint graph, because prior knowledge is typically not cyclic. In general, the SCCs of a constraint graph can be solved in any order due to the global parameter independence property.

The algorithm for solving an SCC of a constraint graph is a straightforward modification of the dynamic programming algorithm described above. Specifically, parent graphs are created for each variable in the SCC but defined only over the union of possible parents for that variable. Consider the case of a simple, four-node cycle with no self-loops such that $W \rightarrow X \rightarrow Y \rightarrow Z \rightarrow W$. A parent graph is defined for each variable in $W \cup X \cup Y \cup Z$ but only over valid parents. For example, the parent graph for $X_1$ would be over only variables in $W$. Then, an order graph is defined with entries that violate the edge structure of the constraint graph filtered out. The first layer of the order graph would be unchanged with only singletons, but the second layer would prohibit entries with two variables from the same layer because there are no valid orderings in which $X_i$ is a parent of $X_j$, and would prohibit entries in which a variable $W$ is joined with a variable of $Y$. One can identify valid entries by taking the entries of a previous layer and iterating over each variable present, adding all valid parents for that variable which are not already present in the set.

A simple example illustrating the algorithm is a constraint graph made up of a four node cycle where each node contains only a single variable (Fig. 2A). The parent graphs defined for this would consist solely of two entries, the null entry and the entry corresponding to the only valid parent. The first layer of the order graph would be all variables as previously (Fig. 2B). However, once a variable is chosen to start the topological ordering the order of the remaining variables is fixed because of the constraints, producing a far simpler lattice.

Because constraint graphs can encode a wide variety of different constraints, the complexity of the task depends on the structure of the constraint graph. Broadly, the results from *Ordyniak & Szeider (2013)* still hold, namely, that acyclic constraint graphs can be solved in quadratic time. As was found in *Fan, Malone & Yuan (2014)*, because each SCC can be solved independently, the time complexity for constraint graphs containing a cycle corresponds to the time complexity of the worst case component. Fortunately, although

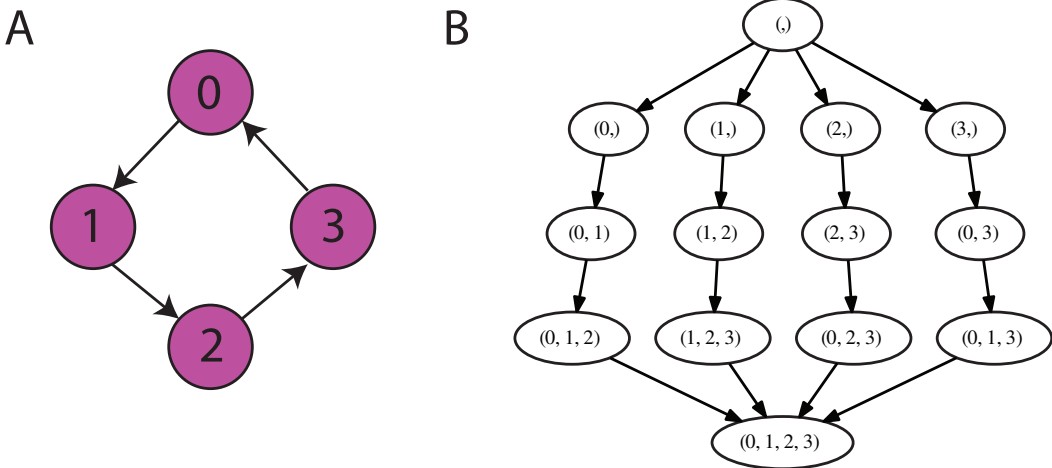

**Figure 2  An example of a constraint graph and resulting order graph.** (A) A constraint graph is defined as a cycle over four nodes with each node containing a single variable. (B) The resulting order graph during the BNSL task. It is significantly sparser than the typical BNSL task because after choosing a variable to start the topological ordering the remaining variables must be added in the order defined by the cycle.

the complexity of a node engaging in a cycle is still exponential, it is only exponential with respect to the number of variables that node interacts with. Adding additional, equally sized nodes to the constraint graph only causes the algorithm to grow linearly in time and has no additional memory cost if the components are solved sequentially.

The algorithm described above has five natural cases and are described below.

**One node, no parents, no self loop:** The variables in this node contain no parents, so nothing needs to be done to find the optimal parent sets given the constraints. This naturally takes $O(1)$ time to solve.

**One node, no parents, self loop:** This is equivalent to exact BNSL with no prior knowledge. In this case, the previously proposed dynamic programming algorithm is used to identify the optimal structure of the subnetwork containing only variables in this node. This takes $O(n2^n)$ time where $n$ is the number of variables in the node.

**One node, one or more parent nodes, no self loop:** In this case it is impossible for a cycle to be formed in the resulting Bayesian network regardless of optimal parent sets, so we can justify solving every variable in this node independently by the global parameter independence property. Doing so results in a significant improvement over applying the algorithm naively because neither the parent graphs nor the order graph need to be explicitly calculated or stored. The optimal parent set can be calculated without the need for dynamic programming because the optimal topological ordering does not need to be discovered. Because no dynamic programming needs to be done, there is no need to store either the parent or order graphs in memory. This takes $O(nm^k)$ time, where $n$ is the number of variables in the node, $m$ is the number of possible parents, and $k$ is the maximum number of parents that a node can have, in this case set by the MDL algorithm. If $k$ is set to any constant value, then this step requires quadratic time with respect to the number of possible parents and linear with respect to the number of variables in the node.

**One node, one or more parents, self loop:** Initially, one may think that solving this SCC could involve taking the union of all variables from all involved nodes, running exact BNSL over the full set, and simply discarding the parent sets learned for the variables not in the currently considered node. However, in the same way that one should not handle prior knowledge by learning the optimal graph over all variables and discarding edges which offend the prior knowledge, one should not do the same in this case. Instead, a modification to the dynamic programming algorithm itself can be made to restrict the parent sets on a variable-by-variable basis. For simplicity, we define the variables in the current node of the constraint graph as $X$ and the union of all variables in the parent nodes in the constraint graph as $Y$. We begin by setting up an order graph, as usual defined over $X$. We then add $Y$ to each node in the order graph such that the root node now is now comprised of $Y$ instead of the empty set and the leaf node is comprised of $X \cup Y$ instead of just $X$. Because the primary purpose of the order graph is to identify the optimal parent sets that do not form cycles, this addition is intuitive because it is impossible to form a cycle by including any of the variables in $Y$ as parents for any of the variables in $X$. In other words, if one attempted to find the optimal topological ordering over $X \cup Y$ it would always begin with the variables in $Y$ but would be invariant to the ordering of $Y$. Parent graphs are then created for all variables in $X$ but are defined over the set of all variables in $X \cup Y$, because that is the full set of parents that the variables could be drawn from. This restriction allows the optimal parents for each variable in $X$ to be identified without wasting time considering what the parent set for variables in $Y$ should be, or potentially throwing away the optimal graph because of improper edges leading from a variable in $Y$ to a variable in $X$. This step takes $O(n2^{n+m})$ time, where $n$ is the number of variables in the node and $m$ is the number of variables in the parent nodes. This is because we only need to define a parent graph for the variables in the node we are currently considering, but these parent graphs must be defined over all variables in the node plus all the variables in the parent nodes.

**Multiple nodes:** The algorithm as presented initially is used to solve an entire component at the same time.

## RESULTS

While it is intuitive how a constraint graph provides computational gains by splitting the structure learning task into subproblems, we have thus far only alluded to the idea that prior knowledge can provide efficiencies past that. In this section we examine the computational gains achieved in the three non-trivial cases of the algorithm presented in 'Solving a component of the constraint graph'.

### Acyclic constraint graphs can model the global stock market

First, we examine the computational benefits of an acyclic constraint graph modeling the global stock market. In particular, we want to identify for each stock which other stocks are predictive to its performance. We chose to do this by learning a Bayesian network over the opening and closing prices of 54 top performing stocks from the New York Stock Exchange (NYSE) in the United States, the Tokyo Stock Exchange (TSE) in Japan, and the Financial Times Stock Exchange (FTSE) in England. Learning a Bayesian network

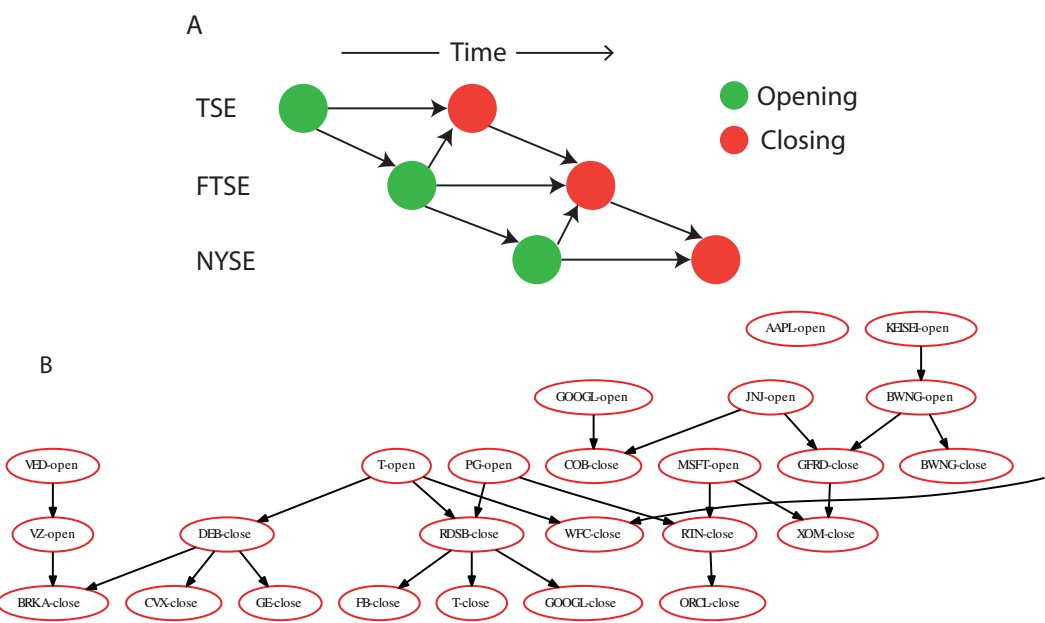

**Figure 3** **A section of the learned Bayesian network of the global stock market.** (A) The constraint graph contains six nodes, the opening and closing prices for each of the three markets. These are connected such that the closing prices in a market depend on the opening prices but also the most recent international activity. (B) The most connected subset of stocks from the learned network covering 25 variables.

over all 108 variables is clearly infeasible, so we encode in our constraint graph some common-sense restrictions (Fig. 3A). Specifically, opening and closing prices for the same market are grouped into separate nodes, for a total of six nodes in the constraint graph. There are no self-loops because the opening price of one stock does not influence the opening price of another stock. Naturally, the closing prices of one group of stocks are influenced by the opening price of the stocks from the same market, but they are also influenced by the opening or closing prices of any markets which opened or closed in the meantime. For instance, the TSE closes after the FTSE opens, so the FTSE opening prices have the opportunity to influence the TSE closing prices. However, the TSE closes before the NYSE opens, so the NYSE cannot influence those stock prices. The dataset consists of opening and closing prices from these stocks between December 2nd, 2015 and November 29th, 2016, binarized to indicate whether the value was an increase compared to the prior price seen.

The resulting Bayesian network has some interesting connections (Fig. 3B). For example, the opening price of Microsoft influences the closing price of Raytheon, and the closing price of Debenhams plc, a British multinational realtor, influences the closing price of GE. In addition, there were some surprising and unexplained connections, such as Google and Johnson & Johnson influencing the closing price of Cobham plc, a British defense firm. Given that this example is primarily to illustrate the types of constraints a constraint graph can easily model, we suggest caution in thinking too deeply about these connections.

**Table 1  Model comparison between naive Bayes, Bayesian network classifiers (BNC), and random forest.** Three algorithms were evaluated on the UCI handwritten digits dataset, fed in the binarized value corresponding to whether the intensity of a pixel was above average. The fitting time and test set accuracy are reported for each algorithm.

| Model | Train time (s) | Test set accuracy |
| --- | --- | --- |
| Naive Bayes | 0.05 | 0.79 |
| BNC | 0.5 | 0.81 |
| Random forest | 1.4 | 0.89 |

It took only ∼35 s on a computer with modest hardware to run BNSL over 250 samples. If we set the maximum number of parents to three, which is the empirically determined maximum number of parents, then it only takes ∼2 s to run. In contrast it would be infeasible to run the exact BNSL algorithm on even half the number of variables considered here.

## Constraint graphs allow learning of Bayesian network classifiers

Bayesian network classifiers are an extension of Bayesian networks to supervised learning tasks by defining a Bayesian network over both the feature variables and the target variables together. Normal inference methods are used to predict the target variables given the observed feature variables. In the case where feature variables are always observed, only the Markov blanket of the target variables must be defined, i.e., their parents and children. The other variables are independent of the target variables and can be discarded, serving as a form of feature selection.

A popular Bayesian network classifier is the naive Bayes classifier that defines a single class variable as the parent to all feature variables. A natural extension to this method is to learn which features are useful, instead of assuming they all are, thereby combining feature selection with parameter learning in a manner that has some similarities to decision trees. This approach can be modeled by using a constraint graph that has all feature variables $X$ in one node and all target variables $y$ in its parent node, such that $y \rightarrow X$.

We empirically evaluated the performance of learning a simple Bayesian network classifier on the UCI Digits Dataset. The digits dataset is a collection of $8 \times 8$ images of handwritten digits, where the features are discretized values between 0 and 16 representing the intensity of that pixel and the labels are between 0 and 9 representing the digit stored there. We learn a Bayesian network where the 64 pixels are in one node in the constraint graph and the class label is by itself it another node in the constraint graph that serves as a parent. We then train a Bayesian network classifier, a naive Bayes classifier, and a random forest classifier comprised of 100 trees, on a test set of 1,500 images and test their performance on a held out 297 images. As expected, the learned Bayesian network classifier falls between naive Bayes and the random forest in terms of both training time and test set performance (Table 1).

Futhermore, more complicated Bayesian network classifiers can be learned with different constraint graphs. One interesting extension is that instead of constraining all features to be children of the target variable, to allow features to be either parents or children of the target variable. This can be specified by a cyclic constraint graph where $y \rightarrow X \rightarrow y$, preventing

**Table 2** **Algorithm comparison on a node with a self loop and other parents.** The exact algorithm and the constrained algorithm proposed here were on a SCC composied of a main node with a self loop and one parent node. Shown are the results of increasing the number of variables in the main node while keeping the variables in the parent node steady at five, and the results of increasing the number of variables in the parent node while keeping the number of variables in the main node constant. For both algorithms we show the number of nodes across all parent graphs (PGN), the number of nodes in the order graph (OGN), the number of edges in the order graph (OGE) and the time to compute.

| | Exact | | | | Constraint graph | | | |
|---|---|---|---|---|---|---|---|---|
| | PGN | OGN | OGE | Time (s) | PGN | OGN | OGE | Time (s) |
| Variables | | | | | | | | |
| 4 | 2,304 | 512 | 2,304 | 0.080 | 1,024 | 16 | 32 | 0.033 |
| 8 | 53,248 | 8,192 | 53,248 | 1.30 | 32,768 | 256 | 1,024 | 0.545 |
| 12 | 1,114,112 | 131,072 | 1,114,112 | 27.03 | 786,432 | 4,096 | 24,576 | 9.56 |
| Parents | | | | | | | | |
| 4 | 2,304 | 512 | 2,304 | 0.087 | 1,280 | 32 | 80 | 0.045 |
| 8 | 53,248 | 8,192 | 53,258 | 1.401 | 20,480 | 32 | 80 | 0.356 |
| 12 | 1,114,112 | 131,072 | 1,114,112 | 27.22 | 327,680 | 32 | 80 | 4.01 |

the model from spending time identifying dependencies between the features. Finally, in cases where some features may be missing, it may be beneficial to model all dependencies between the features in order to allow inference to flow from observed variables not directly connected to the target variables to the target variables. This can be modeled by adding a self loop on the features variables $X$, allowing all edges to be learned except those between pairs of target variables. Learning a Bayesian network classifier in this manner will suffer from the same computational challenges as an unconstrained version, given the looseness of the constraints.

## Self-loops and parents

We then turn to the case where the strongly connected component is a main node with a self loop and a parent node. Because an order graph is defined only over the variables in the main node its size is invariant to the number of variables in the parent node, allowing for speed improvements when it comes to calculating the shortest path. In addition, parent graphs are only defined for variables in the parent set, and so while they are not smaller than the ones in the exact algorithm, there are fewer. We compare the computational time and complexity of the underlying order and parent graphs between the exact algorithm over the full set of variables and the modified algorithm based on a constraint graph (Table 2). The data consisted of randomly generated binary values, because the running time does not depend on the presence of underlying structure in the data. We note that in all cases there are significant speed improvements and simpler graphs but that there are particularly encouraging speed improvements when the number of variables in the main node are increased. This suggests that it is always worth the time to identify which variables can be moved from a node with a self loop to a separate node.

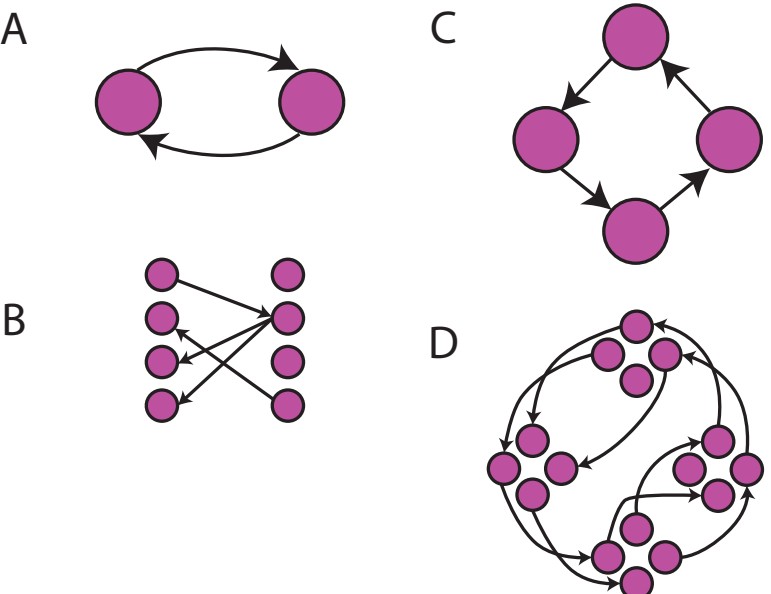

**Figure 4 Cyclic constraint graphs.** (A) This constraint graph is comprised of a simple two node cycle with each node containing four variables. (B) The learned Bayesian network on random data where some variables were forced to identical values. Each circle here corresponds to a variable in the resulting Bayesian network instead of a node in the constraint graph. There were multiple possible cycles which could have been formed but the constraint graph prevented that from occuring. (C) This constraint graph now encodes a four node cycle each with four variables. (D) The learned Bayesian network on random data with two distinct loops of identical values forced. Again, no loops are formed.

## Cyclic constraint graphs

Lastly, we consider constraint graphs that encode cyclic prior knowledge. We visually inspect the results from cyclic constraint graphs to ensure that they do not produce cyclic Bayesian networks even when the potential exists. Two separate constraint graphs are inspected, a two node cycle and a four node cycle (Figs. 4A and 4C). The dataset is comprised of random binary values, where the value of one variable in the cycle is copied to the other variables in the cycle to add synthetic structure. However, by jointly solving all nodes cycles are avoided while dependencies are still captured (Figs. 4B and 4D).

We then compare the exact algorithm without constraints to the use of an appropriate constraint graph in a similer manner as before (Table 3). This is done first for four node cycles where we increase the number of variables in each node of the constraint graph and then for increasing sized cycles with three variables per node. The exact algorithm likely produces structures that are invalid according to the constraints and so this comparison is done solely to highlight that efficiencies are gained by considering the constraints. In each case using a constraint graph yields simpler parent and order graphs and the computational time is significantly reduced. The biggest difference is in the number of nodes in the parent graphs, as the constraints place significant limitations on which variables are allowed to be parents for which other variables. Since the construction of the parent graph is the only part of the algorithm which considers the dataset itself it is unsurprising that significant savings are achieved for larger datasets when much smaller parent graphs are used.

**Table 3   Algorithm comparison on a cyclic constraint graph.** The exact algorithm and the constrained algorithm proposed here were run for four node cycles with differing numbers of variables, cycles with different numbers of nodes but three variables per node, and differing numbers of samples for a four-node, three-variable cycle. All experiments with differing numbers of variables or nodes were run on 1,000 randomly generated samples. Shown for both algorithms are the number of nodes across all parent graphs (PGN), the number of nodes in the order graph (OGN), the number of edges in the order graph (OGE) and the time to compute. Since the number of nodes does not change as a function of samples those values are not repeated in the blank cells.

| | Exact | | | | Exact | | | |
|---|---|---|---|---|---|---|---|---|
| | PGN | OGN | OGE | Time (s) | PGN | OGN | OGE | Time (s) |
| Variables | | | | | | | | |
| 1 | 32 | 16 | 32 | 0.005 | 8 | 14 | 16 | 0.005 |
| 2 | 1,024 | 256 | 1,024 | 0.036 | 32 | 186 | 544 | 0.014 |
| 3 | 24,576 | 4,096 | 24,576 | 0.611 | 96 | 3,086 | 16,032 | 0.320 |
| 4 | 524,288 | 65,536 | 525,288 | 14.0 | 256 | 54,482 | 407,328 | 7.12 |
| Nodes | | | | | | | | |
| 2 | 192 | 64 | 192 | 0.111 | 48 | 56 | 150 | 0.008 |
| 4 | 24,576 | 4,096 | 24,576 | 0.634 | 96 | 3,086 | 16,032 | 0.217 |
| 6 | 2,359,296 | 262,144 | 2,359,296 | 60.9 | 144 | 168,068 | 1,307,358 | 26.12 |
| Samples | | | | | | | | |
| 100 | 24,576 | 4,096 | 24,576 | 0.357 | 96 | 3,086 | 16,032 | 0.311 |
| 1,000 | – | – | – | 0.615 | – | – | – | 0.211 |
| 10,000 | – | – | – | 2.670 | – | – | – | 0.357 |
| 100,000 | – | – | – | 243.9 | – | – | – | 10.41 |

## DISCUSSION

Constraint graphs are a flexible way of encoding into the BNSL task prior knowledge concerning the relationships among variables. The graph structure can be exploited to identify potentially massive computational gains, and acyclic constraint graphs make problems tractable which would be infeasible to solve without constraints. This is particularly useful in cases where there are both a great number of variables and many constraints present from prior knowledge. We anticipate that the automatic manner in which parallelizable subtasks are identified in a constraint graph will be of particular interest given the recent increase in availability of distributed computing.

Although the networks learned in this paper are discrete, the same principles can be applied to all types of Bayesian networks. Because the constraint graph represents only a restriction in the parent set on a variable-by-variable basis, the same algorithms that are used to learn linear Gaussian or hybrid networks can be seamlessly combined with the idea of a constraint graph. In addition, most of the approximation algorithms which have been developed for BNSL can be modified to take into account constraints because these algorithms simply encode a limitation on the parent set for each variable.

One could extend constraint graphs in several interesting ways. The first is to assign weights to edges so that the weight represents the prior probability that the variables in the parent set are parents of the variables in the child set, perhaps as pseudocounts to take into account when coupled with a Bayesian scoring function. A second way is to incorporate "hidden nodes" that are variables which model underlying, onobserved phenomena and

can be used to reduce the parameterization of the network. Several algorithms have been proposed for learning the structure of a Bayesian network given hidden variables (*Elidan et al., 2001*; *Elidan & Friedman, 2005*; *Friedman, 1997*). Modifying these algorithms to obey a constraint graph seems like a promising way to incorporate restrictions on this difficult task. A final way may be to encode ancestral relationships instead of direct parent relationships, indicating that a given variable must occur at some point before some other variable in the topological ordering.

## ACKNOWLEDGEMENTS

We would like to acknowledge Maxwell Libbrecht, Scott Lundberg, and Brandon Malone for many useful discussions and comments on drafts of the paper.

### Funding

This work is supported by an NSF IGERT grant DGE-1258485. There was no additional external funding received for this study. The funders had no role in study design, data collection and analysis, decision to publish, or preparation of the manuscript.

### Grant Disclosures

The following grant information was disclosed by the authors:
NSF IGERT: DGE-1258485.

### Competing Interests

The authors declare there are no competing interests.

### Author Contributions

- Jacob M. Schreiber conceived and designed the experiments, performed the experiments, analyzed the data, wrote the paper, prepared figures and/or tables, performed the computation work, reviewed drafts of the paper.
- William S. Noble analyzed the data, wrote the paper, reviewed drafts of the paper.

### Data Availability

  Code implementing the concept is available at GitHub:
  www.github.com/jmschrei/pomegranate
  Data and code reproducing the figures are available at GitHub: https://github.com/jmschrei/constraint_graphs.

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
