# Peer review of "Finding the optimal Bayesian network given a constraint graph"

_PeerJ Computer Science, doi:10.7717/peerj-cs.122_

## Round 0.1 · original submission · Minor Revisions

· Academic Editor

Minor Revisions

Your manuscript has been reviewed by two of our referees.
Comments from their reports appear below. When you resubmit your manuscript, please include a summary of the changes made and a brief response to all recommendations and criticisms.

·

Basic reporting

The authors propose the notion of constraint graphs for the task of handling prior knowledge in a Bayesian network, and give indications on the efficiency gains over existing methods. They cite Perrier et al 2014 in which the same task is accomplished by a different method. It must be noted that Perrier et al give explicitly the complexity of the method. Contrastingly, this manuscript begins by calling the new method 'intuitive', and does not mention the complexity, except for the tables and some statements which implicitly say that it is polynomial (46 to 52).

In my view, for the work in this manuscript to be complete, the complexity issue is better to be addressed bit more analytically. It would not be very difficult as the method in this manuscript basically relies on Tarjan's algorithm (139 to 143).

Experimental design

No comments

Validity of the findings

The tables in the results section give considerable evidence for the authors' claim. It would be better to include some stronger evidence. Or the authors may verify it analytically.

Reviewer 2 ·

Basic reporting

* The article fulfills the criteria of PeerJ, except maybe self-containedness: I'd propose that the authors shortly review the usually applied scores for Bayesian network learning, i.e. MDL, BIC, BD, and mention their property that they decompose over the network's nodes.

* Also, I think the dynamic programming method by Malone et al., 2011 should be described in a more detailed way. I see that any algorithm (even a greedy one) could be used, as long as it obeys the constrained graph. However, since the authors assume this particular algorithm, I think explaining it in a more detailed way would make the paper more self-contained and understandable for the non-expert.

* Minor comment: Figure 3 doesn't have a sub-figure C, but the caption refers to it (and it actually should refer to B)

Experimental design

* Data-sets should be introduced.
In Section 4.1, stock market data is used, but the source of this data is not given.
In Section 4.2, the data set is not described at all (synthetic?).
In Section 4.3, the data seems to be random, except that certain variables are clamped to the same values; please describe this process in a more detailed way.

Validity of the findings

Findings are valid. Some more experiments on real data would be appropriate.

Additional comments

The paper is surely somewhat incremental, in light of Fan et al. and Malone et al. However, since novelty is not a primary issue for PeerJ, this should not be a concern here. I find the constraint graph elegant, easy to understand and use for non-experts (in probabilistic modeling), and that the approach interplays nicely with the "spirit" of Bayesian networks (combining expert knowledge and data).

Generally, it would be desirable to see some more experiments with real-world data.

It would also be interesting to see how large datasets can get (in terms of number of variables), when placing stringent but reasonable constraints on the structure.

---

## Round 0.2 · Minor Revisions

· Academic Editor

Minor Revisions

I would recommend publication of your paper after the remaining (very) minor revisions following the referees' comments.

Reviewer 2 ·

Basic reporting

The authors have properly addressed my concern about self-containedness.
Fig. 3 still contains the minor mistake, but it can easily be fixed without further review.

Experimental design

The data-sets have been described in more detail, so the authors have addressed my concern.

Validity of the findings

The authors have included a small real-world example (feature selection using naive Bayes) which reinforces the validity of their findings.

Additional comments

The authors did a good job in revising the manuscript and I suggest acceptance.

In the final version, please fix the following really minor issues:
* Figure 3 still has a caption mistake
* The Markov Blanket of a BN (defined on page) also includes the co-parents of a node's children.
* On page 8, line 307, grammar: probably "that" needs to be deleted.

---

## Author Rebuttal · Round 0.2

We would like to thank the editor and the reviewers for their helpful comments. We have made several changes to our manuscript to improve its quality that hopefully address their concerns. We have responded individually to their suggestions below. In our response, purple text highlights our response, blue text corresponds to unchanged text from the manuscript, and red text corresponds to changed text from the manuscript.

# Reviewer 1 (Anuradha Mahasinghe)

## Basic reporting

The authors propose the notion of constraint graphs for the task of handling prior knowledge in a Bayesian network, and give indications on the efficiency gains over existing methods. They cite Perrier et al 2014 in which the same task is accomplished by a different method.

Perrier et al consider constraints in the form of an undirected super structure, whereas the constraints in a constraint graph are directed. Thus, although the two tasks are related, they are not the same. Much of the focus in the Perrier et al. paper is how to assign directionality when the constraints themselves do not have direction.

It must be noted that Perrier et al give explicitly the complexity of the method. Contrastingly, this manuscript begins by calling the new method 'intuitive', and does not mention the complexity, except for the tables and some statements which implicitly say that it is polynomial (46 to 52).

In my view, for the work in this manuscript to be complete, the complexity issue is better to be addressed bit more analytically. It would not be very difficult as the method in this manuscript basically relies on Tarjan's algorithm (139 to 143).

We have added text to address the complexity issue further. See the below text, where blue corresponds to previous text and red corresponds to our additions discussing the complexity:

Because constraint graphs can encode a wide variety of different constraints, the complexity of the task depends on the structure of the constraint graph. Broadly, the results from \citet{ordyniak:parameterized} still hold, namely, that acyclic constraint graphs can be solved in quadratic time. As was found in \citet{fan:finding}, because each SCC can be solved independently, the time complexity for constraint graphs containing a cycle corresponds to the time complexity of the worst case component. Fortunately, although the complexity of a node engaging in a cycle is still exponential, it is only exponential with respect to the number of variables that node interacts with. Adding additional, equally sized nodes to the constraint graph only causes the algorithm to grow linearly in time and has no additional memory cost if the components are solved sequentially.

The algorithm described above has five natural cases and are described below.

**One node, no parents, no self loop:** The variables in this node contain no parents, so nothing needs to be done to find the optimal parent sets given the constraints. This naturally takes O(1) time to solve.

**One node, no parents, self loop:** This is equivalent to exact BNSL with no prior knowledge. In this case, the previously proposed dynamic programming algorithm is used to identify the optimal structure of the subnetwork containing only variables in this node. This takes O($n2^{n}$) time where $n$ is the number of variables in the node.

**One node, one or more parent nodes, no self loop:** In this case it is impossible for a cycle to be formed in the resulting Bayesian network regardless of optimal parent sets, so we can justify solving every variable in this node independently by the global parameter independence property. Doing so results in a significant improvement over applying the algorithm naively, because neither the parent graphs nor the order graph need to be explicitly calculated or stored. The optimal parent set can be calculated without the need for dynamic programming because the optimal topological ordering does not need to be discovered. Because no dynamic programming needs to be done, there is no need to store either the parent or order graphs in memory. This takes O($nm^{k}$) time, where $n$ is the number of variables in the node, $m$ is the number of possible parents, and $k$ is the maximum number of parents that a node can have, in this case set by the MDL algorithm. If $k$ is set to any constant value, then this step requires quadratic time with respect to the number of possible parents and linear with respect to the number of variables in the node.

**One node, one or more parents, self loop:** Initially, one may think that solving this SCC could involve taking the union of all variables from all involved nodes, running exact BNSL over the full set, and simply discarding the parent sets learned for the variables not in the currently considered node. However, in the same way that one should not handle prior knowledge by learning the optimal graph over all variables and discarding edges which offend the prior knowledge, one should not do the same in this case. Instead, a modification to the dynamic programming algorithm itself can be made to restrict the parent sets on a variable-by-variable basis. For simplicity, we define the variables in the current node of the constraint graph as $X$ and the union of all variables in the parent nodes in the constraint graph as $Y$. We begin by setting up an order graph, as usual defined over $X$. We then add $Y$ to each node in the order graph such that the root node now is now comprised of $Y$ instead of the empty set and the leaf node is comprised of $X \cup Y$ instead of just $X$. Because the primary purpose of the order graph is to identify the optimal parent sets that do not form cycles, this addition is intuitive because it is impossible to form a cycle by including any of the variables in $Y$ as parents for any of the variables in $X$. In other words, if one attempted to find the optimal topological ordering over $X \cup Y$ it would always begin with the variables in $Y$ but would be invariant to the ordering of $Y$. Parent graphs are then created for all variables in $X$ but are defined over the set of all variables in $X \cup Y$, because that is the full set of parents that the variables could be drawn from. This restriction allows the optimal parents for each variable

in $X$ to be identified without wasting time considering what the parent set for variables in $Y$ should be, or potentially throwing away the optimal graph because of improper edges leading from a variable in $Y$ to a variable in $X$. This step takes O($n2^{n+m}$) time, where $n$ is the number of variables in the node and $m$ is the number of variables in the parent nodes. This is because we only need to define a parent graph for the variables in the node we are currently considering, but these parent graphs must be defined over all variables in the node plus all the variables in the parent nodes.

## Experimental design

No comments

## Validity of the findings

The tables in the results section give considerable evidence for the authors' claim. It would be better to include some stronger evidence. Or the authors may verify it analytically.

In addition to discussing the complexity further we have added in another example of learning a Bayesian network classifier on the UCI digits dataset. Please see below where we have copied the full section.

# Reviewer 2 (Anonymous)

## Basic reporting

\* The article fulfills the criteria of PeerJ, except maybe self-containedness: I'd propose that the authors shortly review the usually applied scores for Bayesian network learning, i.e. MDL, BIC, BD, and mention their property that they decompose over the network's nodes.

We have added a section describing the score functions for BNSL, mentioning that they are decomposable. The text is below:

Structure learning is flexible with respect to the score function used to identify the optimal graph. There are many score functions that typically aim to penalize the log likelihood of the data by the complexity of the graph to encourage sparser structures. These usually come in the form of Bayesian score functions, such as Bayesian-Dirichlet \cite{FIXME}, or those derived from information theory, such as minimum description length (MDL) \cite{FIXME}. Most score functions decompose across variables of a Bayesian network according to the global parameter independence property, such that the score for a dataset given a model is equal to the product of the score of each variable given its parents. While constraint graphs remain agnostic to the specific score function used, we assume that MDL is used as it has several desirable computational benefits. For review, MDL defines the score as the following:
\begin{equation}
    MDL(D|M) = P(D|M) - \frac{1}{2}\log(N)|B|
\end{equation}
where $|B|$ defines the number of parameters in the network. The term ``minimum description length'' arises from needing $\frac{1}{2}\log(N)$ bits to represent each parameter in the model, making the second term the total number of bits needed to represent the full model. The MDL score function has the convenient property that a variable cannot have more than $\log\left(\frac{n}{\log(n)}\right)$ parents given $n$ samples, greatly reducing computational time.

\* Also, I think the dynamic programming method by Malone et al., 2011 should be described in a more detailed way. I see that any algorithm (even a greedy one) could be used, as long as it obeys the constrained graph. However, since the authors assume this particular algorithm, I think explaining it in a more detailed way would make the paper more self-contained and understandable for the non-expert.

We have added more details to our explanation of the algorithm, as well as clarified further that any algorithm can be used. The full paragraph now reads as follows (with untouched text in blue and modified text in red):

Although solving a component in a constraint graph can be accomplished by a variety of algorithms including heuristic algorithms, we assume for this paper that one is using some variant of the exact dynamic programming algorithm proposed by \citet{malone:memory}. We briefly review that algorithm here.

The goal of the algorithm is to identify the optimal Bayesian network defined over the set of variables without having to repeat any calculations and without having to use excessive

memory. This is done by defining additional graphs, the parent graphs and the order graph. We will refer to each node in these graphs as ``entries'' to distinguish them from the constraint graph and the learned Bayesian network. A parent graph is defined for each variable and can be defined as a lattice, where the entries to some layer $i$ correspond to combinations of all other variables of size $i$. Each entry is connected to the entries in the previous layers that are subsets of that entry such that $(\{X_{1}, X_{2}\})$ would be connected to both $X_{1}$ and $X_{2}$. For each entry, the score of the variable is calculated using the parents in the entry and compared to the scores held in the parent entries, recording only the best scoring value and parent set amongst them. These entries then hold the dynamically calculated best parent set and associated score, allowing for constant time lookups later on of the best parent set given a set of possible parents. The order graph is structured in the same manner as the parent graphs except over all variables. In contrast with the parent graphs, it is the edges that store useful information in the form of the score associated with adding a given variable to the set of seen variables stored in the entry and the parent set that yields this score. Each path from the empty root node to the leaf node containing the full set of variables encodes the optimal network given a topological sort of the variables, and the shortest path encodes the optimal network. This data structure reduces the time required to find the optimal Bayesian network from $O(n2^{n(n-1)})$ time in the number of variables to $O(n2^{n})$ time in the number of variables without the need to keep a large cache of values.

* Minor comment: Figure 3 doesn't have a sub-figure C, but the caption refers to it (and it actually should refer to B)

We have corrected this mistake.

## Experimental design

* Data-sets should be introduced.
In Section 4.1, stock market data is used, but the source of this data is not given.

We have added the following clarifying text:

The dataset consists of opening and closing prices from these stocks between December 2nd 2015 and November 29th 2016, binarized to indicate whether the value was an increase compared to the prior price seen.

In Section 4.2, the data set is not described at all (synthetic?).

We have added the following clarifying text:

The data consisted of randomly generated binary values, because the running time does not depend on the presence of underlying structure in the data.

In Section 4.3, the data seems to be random, except that certain variables are clamped to the same values; please describe this process in a more detailed way.

We have added in the following clarifying text:

The dataset is comprised of random binary values, where the value of one variable in the cycle is copied to the other variables in the cycle to add synthetic structure.

## Validity of the findings

Findings are valid. Some more experiments on real data would be appropriate.

Please see our comment below.

## Comments for the Author

The paper is surely somewhat incremental, in light of Fan et al. and Malone et al. However, since novelty is not a primary issue for PeerJ, this should not be a concern here. I find the constraint graph elegant, easy to understand and use for non-experts (in probabilistic modeling), and that the approach interplays nicely with the "spirit" of Bayesian networks (combining expert knowledge and data).

Generally, it would be desirable to see some more experiments with real-world data.

It would also be interesting to see how large datasets can get (in terms of number of variables), when placing stringent but reasonable constraints on the structure.

We have added other examples that we believe can help highlight the effectiveness of constraint graphs. We first describe how a constraint graph can be used to learn a variety of Bayesian network classifiers with an example of learning one on the UCI digits dataset. We then show that the learned Bayesian network classifier falls between naive Bayes and a random forest in terms of both accuracy and speed, as one would expect from an efficient implementation. Please see section 4.2 and table 2 for the results. We have copied the section in full for the reviewer:

**4.2 Constraint Graphs Allow Learning of Bayesian Network Classifiers**

Bayesian network classifiers are an extension of Bayesian networks to supervised learning tasks by defining a Bayesian network over both the feature variables and the target variables together. Normal inference methods are used to predict the target variables given the observed feature variables. In the case where feature variables are always observed, only the Markov

blanket of the target variables must be defined, i.e. their parents and children. The other variables are independent of the target variables and can be discarded, serving as a form of feature selection.

A popular Bayesian network classifier is the naive Bayes classifier that defines a single class variable as the parent to all feature variables. A natural extension to this method is to learn which features are useful, instead of assuming they all are, thereby combining feature selection with parameter learning in a manner that has some similarities to decision trees. This approach can be modeled by using a constraint graph that has all feature variables $X$ in one node and all target variables $y$ in its parent node, such that $y \rightarrow X$.

We empirically evaluated the performance of learning a simple Bayesian network classifier on the UCI Digits Dataset. The digits dataset is a collection of 8x8 images of handwritten digits, where the features are discretized values between 0 and 16 representing the intensity of that pixel and the labels are between 0 and 9 representing the digit stored there. We learn a Bayesian network, where the 64 pixels are in one node in the constraint graph and the class label is by itself it another node in the constraint graph that serves as a parent. We then train a Bayesian network classifier, a naive Bayes classifier, and a random forest classifier on a test set of 1500 images and test their performance on a held out 297 images. As expected, the learned Bayesian network classifier falls between naive Bayes and the random forest in terms of both training time and test set performance (Table.~\ref{table:bayesnetclassifier}).

Futhermore, more complicated Bayesian network classifiers that can be learned with different constraint graphs. One interesting extension is that instead of constraining all features to be children of the target variable, to allow features to be either parents or children of the target variable. This corresponds to learning the full Markov blanket over the target variables. This can be specified by a cyclic constraint graph where $y \rightarrow X \rightarrow y$, preventing the model from spending time identifying dependencies between the features. Finally, in cases where some features may be missing, it may be beneficial to model all dependencies between the features in order to allow inference to flow from observed variables not in the Markov blanket through unobserved variable in the Markov blanket. This can be modeled by adding a self loop on the features variables $X$, allowing all edges to be learned except those between pairs of target variables. Learning a Bayesian network classifier in this manner will suffer from the same computational challenges as an unconstrained version, given the looseness of the constraints.

---

## Round 0.3 · accepted · Accept

· Academic Editor

Accept

I am writing to inform you that your manuscript - Finding the optimal bayesian network given a constraint graph - has been Accepted for publication.